# A Roof Refurbishment Strategy to Improve the Sustainability of Building Stock: A Case Study

**María J. Ruá \*** , **Ángel M. Pitarch** , **Inés Arín** **and Lucía Reig**

Mechanical Engineering and Construction Department, Universitat Jaume I, 12071 Castellón de la Plana, Spain; pitarcha@uji.es (Á.M.P.); arin@uji.es (I.A.); lreig@uji.es (L.R.)
\* Correspondence: rua@uji.es

**Abstract:** The aging of the building stock in most cities highlights the relevance of refurbishment to achieve sustainability. Current refurbishment practices are often short-sighted and do not encompass holistic strategies beyond energy saving. This research study aims to analyze the factors involved in roof refurbishment versus current decision-making determinants. The objective is to identify the barriers that hinder their implementation and to find arguments to support roof renovations. A multicriteria analysis, which considered environmental, economic and performance factors, was employed to select optimal roof refurbishment solutions. This study evaluated five solutions. With interviews held with construction professionals and a survey of experts and homeowners, the preferences and criteria for making decisions about roof refurbishments were analyzed. Simulation tools were then used to estimate the energy savings, payback periods and environmental impact for a representative building in the study area. The results were extrapolated to a neighborhood level. The results highlight the importance of considering factors, such as weight, cost and user preferences when selecting suitable refurbishment solutions. The findings not only estimate the potential energy savings and carbon emission reductions in the area but also underscore the relevance of roof refurbishments for prolonging a building's life span to contribute to sustainability.

**Keywords:** urban sustainability; roof refurbishment; multicriteria analysis; simulation tools; energy performance

## 1. Introduction

Energy efficiency in buildings has been a key objective in recent years because of its contribution to low-carbon economies. In 2018, buildings accounted for more than 40% of Europe's energy use. Buildings have been put forward as an emission reduction target alongside other sectors, such as transport, agriculture, waste and industry. As part of the European Green Deal strategy, European Union members agreed to increase the greenhouse gas emission reduction target for 2030 from 40% to 55% by means of the "Fit for 55" initiative. This commitment is set out in the Climate Law Regulation and is, thus, becoming a requisite. To reach the new target, the European Commission presented a package of legislative proposals to revise and update EU legislation on land use, taxation, transport and energy to ensure that the EU achieves climate neutrality by 2050. Regarding buildings, the Council and the Parliament reached a provisional political agreement on a proposal to revise the Energy Performance of Buildings Directive (EPBD) from 2010 [1]. The revised directive sets more ambitious energy performance requirements for new and renovated buildings in the EU, and promotes the building stock's renovation. The main objectives of this revision are that all new buildings should be zero-emission by 2030, and the existing building stock should be transformed into zero-emission buildings by 2050. Zero-emission buildings are defined in Article 2 as buildings with very high energy performance using a small amount of energy that is still needed and fully covered by renewable energies. These buildings will set a new standard for new constructions. Major

renovations will need to reach this level as of 2030, and the entire stock must comply with it by 2050. Article 2 also clarifies that "nearly zero-energy buildings" remain the standard for new buildings until the application of the zero-emission building standard and will be the level to be met by profound renovations until 2030.

When looking at energy classes, according to the data provided by the European Union (EU), 51% of the existing residential building stock is in an energy class below Class D (Classes E, F and G when on the A–G scale), and only 3% of this stock is in energy Class A. Currently, the energy refurbishment rate of the stock in Europe is 1% per year, which is below the rate of 3% recommended by the European Commission [2].

In Spain, the situation is significantly worse because 81% of the building stock is in an energy class below Class D, and only 0.3% is in Class A. The National Energy and Climate Plan aims to increase this rate to 1.2% by 2030, with progressive increases over the years. This is because the refurbishment rate in Spain is 0.1%, which is still much lower than the European average of 1% [3], and is mainly due to the predominant property regime in Spain being home ownership, which makes it very difficult to renovate buildings where several owners with different situations and perspectives co-exist. This often makes it difficult to reach agreements to carry out renovation interventions. Many buildings in Spain are multifamily homes with flat roofs and are typically located in temperate climate zones. Flat roofs are systematically seen in most buildings that were built before 1979, when the first regulation on thermal conditions was approved in Spain.

Refurbishment interventions on roofs usually contribute poorly to overall energy performance because a roof accounts for a low percentage of the thermal envelope. So, it is quite possible that intervention on the roof will not yield the most cost-efficient results. However, as highlighted by Morgado et al. [4], beyond energy improvement, a proper roof is crucial to maintaining the whole building and prolonging its service life because these interventions can prevent diseases that derive from construction deterioration. The proposal for revising the EPBD focuses on reducing operational greenhouse gas emissions [1]. However, initial measures are being taken to address carbon emissions throughout a building's entire life cycle. This highlights the importance of roof refurbishment in enhancing a building's durability and overall performance. Furthermore, given the high cost of the investments needed to adapt buildings' energy performance to currently required standards, roof interventions would be aligned with the philosophy of the Electronic Building Book, whose purpose is to program buildings' partial renovations with realistic, affordable and adequate planning, rather than incurring the excessive cost inherent to comprehensive interventions [5,6]. The proposal to update the EPBD presents "staged renovation" as a solution to the high upfront costs, which may act as an obstacle when renovating "in one go". Moreover, such renovation needs to be thoroughly planned to avoid a situation in which one stage excludes the following stages. The Renovation Passport has been suggested as a voluntary tool for owners and investors, and it provides a roadmap for planned renovation. However, according to current conditions, interventions on roofs, which are not a very noticeable element of the thermal envelope, are not perceived as a priority, and maintenance works are carried out only when a disease or problem is detected. Additionally, the most economical intervention to solve the problem in the short term is almost always selected.

This research study focuses on the potential contribution of roof refurbishment as the most exposed part of the thermal envelope. This study was conducted on a multifamily building with a flat roof that was built before 1979 in a medium-sized Spanish Mediterranean city (Castellón de la Plana). This pattern is probably similar in many cities and municipalities in Southern Europe and North Africa [7]. The first stage of this work analyzed the most commonly used roof restoration solutions based on a multicriteria analysis (environmental, economic and performance) and selected the most appropriate ones. The second phase of this work aimed to quantify the area of different roof typologies and the potential improvement that renovation would entail in a neighborhood located on the coastline of Castellón de la Plana (East Spain), as presented in Figure 1. For this purpose,

the energy savings of some roof rehabilitation solutions were estimated, and the theoretical savings for a statistically representative building of the area, with no insulation on the roof, were determined depending on the refurbishment roof solutions.

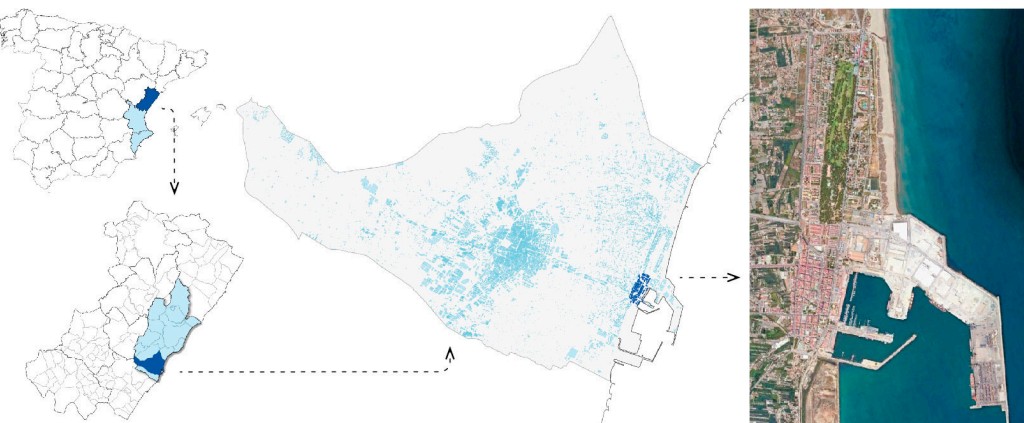

**Figure 1.** Location of the case study neighborhood: Grao de Castellón.

## 2. Materials and Methods

This research was conducted in two stages. Figure 2 summarizes the objectives, methods and main results for each stage in accordance with the way the paper is organized. This study focused on flat roofs without thermal insulation, typically used in cities with a temperate climate in Spain. Then, suitable construction solutions for flat roof refurbishment were analyzed by considering current regulatory standards. Five different refurbishment solutions were selected. Three were walkable roof solutions (inverted with raised paving, inverted with adhered paving and inverted with permeable paving (WINVR, WINVAD and WPER, respectively)); two were non-walkable (green roofs and roofs with gravel protection (NWGRE and NWGRA, respectively)).

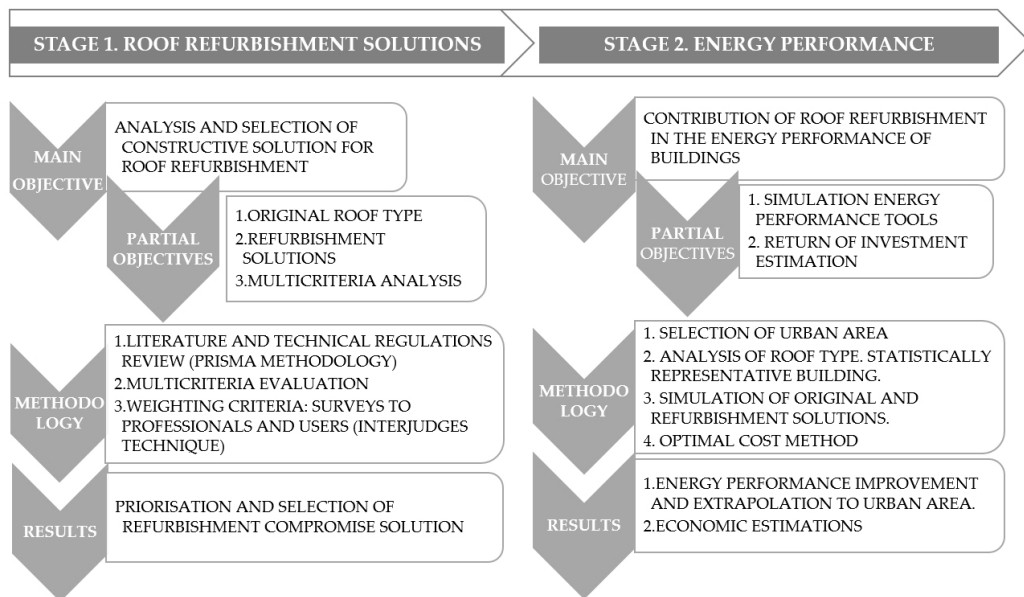

**Figure 2.** Study stages, followed methodologies and main findings.

The multicriteria analysis used to evaluate these five solutions was based on three criteria categories, namely, environmental (A), economic (E) and performance (P), with nine total evaluation indicators. To properly define the indicators to be considered, two population focus groups—one made up of experts in construction and refurbishment and

the other of users/owners of buildings—were asked to answer a survey to analyze their preferences in the decision-making process for building refurbishment and to determine the weighted importance assigned to the different factors. The information provided by planners and builders on the renovation solutions commonly used in actual professional practice also allowed the practical and theoretical solutions to be compared.

The multicriteria analysis established an order of preference for flat roof refurbishment solutions, from the most suitable to the least suitable. Additionally, some suggestions were collected from the obtained results.

In the second study stage, three of the initially proposed solutions were further analyzed to estimate the energy performance of roofs. To do so, simulation values were obtained employing an energy certification tool that requires modeling a complete building. For this reason, a statistically representative building in the urban area that presented the typical non-insulated flat roof solution was selected. This case study selection was based on an exhaustive and detailed study about building typologies, roof types, roof surfaces, etc. This building was simulated with a tool officially approved by the Ministry of Energy certification, and the improvement made with the chosen refurbishment solutions was subsequently tested. The obtained results were used to estimate the potential refurbishment at the neighborhood level by extrapolating to the total area of this roof type. Finally, based on the acquired results, the payback periods for the considered rehabilitation interventions were estimated.

## 3. Stage 1 Results: Analysis of Refurbishment Solutions

### 3.1. Thermal Insulation of Roofs in the Current Building Stock

Roof types in Mediterranean cities can be classified according to their thermal insulation. In line with this, the first mandatory regulation in Spain with requirements for thermal conditions for buildings dates back to 1979: the Basic Housing Regulation on Thermal Conditions [8]. Therefore, in mild climate zones in Spain, buildings before that year present no insulated roofs at all. After the 1979 regulation, insulation was required, but it was not until 2006 when the Building Technical Code (CTE, in Spanish), updating previous and obsolete regulations in the building sector, established more ambitious energy restrictions. The part of the CTE that regulates the thermal conditions of buildings is called CTE-HE, and has been updated in the last few years and aligned with the European EPBD [9].

Regarding the building typology classification, the research work previously undertaken by the Valencian Government (East Spain) and the Valencian Building Institute IVE, which has been integrated into the European project Tabula, presented a catalog for the building typologies in different climates in Spain [10]. It characterized the thermal envelope depending on climate, building typology and year of construction in Spain. As described in this study, buildings' estimated thermal transmittance has progressively reduced over the years, which has allowed buildings to be organized into periods according to building regulations: "before 1939", "1937–1959", "1960–1979", "1980–2006" and "after 2006". These periods correspond to a reduction in their roof thermal transmittances from 3.08, 1.67, 1.61, 0.56 to 0.45 W/m$^2$K, respectively. The research undertaken by Braulio [11] characterized the building thermal envelope of building stock in the Mediterranean climate over a similar time interval. The results show that thermal envelopes can be characterized depending on year of construction due to regulations on thermal conditions at the time and the use of standard constructive systems.

### 3.2. Identification of Refurbishment Solutions

Multifamily buildings with flat roofs built before 1979, when the first thermal regulation on buildings came into force, were selected as the reference roof to be refurbished. Table 1 summarizes the main roof system layers used as a reference and the five proposed refurbishment solutions, as described in Section 2. These refurbishment systems have been analyzed in previous studies and the corresponding references are included in Table 1. As the first row shows, the reference system presents a roof with no type of thermal insulation.

**Table 1.** Flat roof type to be refurbished and selected roof refurbishment solutions.

| Roof Type | Layers on the Roof | Refs. |
|---|---|---|
| Existent | 1. Ceramic tiles<br>2. Layer of mortar<br>3. Waterproofing membrane<br>4. Lightweight concrete (Slope formations) | [10,11] |
| NWGRE Green roof | 1. Vegetation<br>2. Substrate<br>3. Filtering layer<br>4. Drainage layer<br>5. Protective layer<br>6. Anti-rooting layer<br>7. Waterproofing membrane (EPDM) | [12–16] |
| NWGRA Gravel protection | 1. Gravel<br>2. Separation and drainage layer (geotextile)<br>3. Thermal insulation (XPS)<br>4. Separation and drainage layer (geotextile)<br>5. Waterproofing membrane (EPDM) | [17,18] |
| WINVR Raised paving | 1. Ceramic tiles<br>2. Polypropylene plots<br>3. Separation and drainage layer (geotextile)<br>4. Thermal insulation (XPS)<br>5. Separation and drainage layer (geotextile)<br>6. Waterproofing membrane (EPDM) | [18,19] |
| WINVAD Adhered paving | 1. Ceramic tiles + Cementitious adhesive<br>2. Regularization mortar<br>3. Separation and drainage layer (geotextile)<br>4. Thermal insulation (XPS)<br>5. Separation and draining layer (geotextile) | [18] |
| WPER Permeable paving | 1. Ceramic tiles + thermal insulator<br>2. Separation and drainage layer (geotextile)<br>3. Waterproofing membrane (EPDM) | [20,21] |

### 3.3. Flat Roof Refurbishment Solutions

3.3.1. Definition of Quantitative and Qualitative Criteria for Multicriteria Analysis

The suitability of different flat roof refurbishment solutions depends on several evaluation criteria. In some cases, they are decisive for selection purposes, such as the weight that the pre-existing building structure can support, but are of less relevance in other cases, such as the aesthetic aspect of the finish. Table 2 shows the indicators proposed in this

multicriteria study. Each indicator was rated on a scale from 1 to 5, with 1 being the most unfavorable and 5 the most favorable. Some indicators were qualitatively evaluated based on their technical characteristics, advantages and disadvantages observed in practice, while others were quantitatively assessed by means of measurable variables, whose values were subsequently standardized on a scale from 1 to 5. As the three variables were inverse to the normalized scale (i.e., the heavier the weight, the worse the value), their normalized values were obtained using Equation (1).

$$VNi = 5 \times (Vmin/Vi), \tag{1}$$

where:

VNi is the normalized value of the indicator for constructive solution i;
Vi is the unnormalized value of solution i;
Vmin is the minimum value to be reached of the i solutions.

**Table 2.** Categories and indicators for the multicriteria analysis.

| Category | Indicator | Type of Indicator: Criteria |
|---|---|---|
| A | A.1. Thermal insulation—energy savings | Quantitative: the normalized value of thermal transmittance (W/m$^2$K). |
| | A.2. Recovery—Recycling | Qualitative: the recovering, reusing and recycling potentials of the materials used in rehabilitation. |
| E | E.1. Initial investment cost | Quantitative: the normalized unit cost of executing refurbishment solutions (EUR/m$^2$). |
| | E.2. Maintenance (durability–cost–periodicity) | Qualitative: the durability and the need for maintenance operations in frequency and cost terms. |
| P | P.1. Ease of execution | Quantitative: the system's normalized weight (kN/m$^2$). |
| | P.2. Acoustic insulation | Qualitative: capacity to prevent roof leaks. |
| | P.3. Weight of the system | Qualitative: the roof's aesthetic value. |
| | P.4. Waterproofing—sealing | Quantitative: the system's normalized weight (kN/m$^2$). |
| | P.5. Aesthetic | Qualitative: capacity to prevent roof leaks. |

Regarding the quantitative indicators, Figure 3 summarizes the thermal transmittance, weight and economic cost of the different analyzed refurbishment solutions. For thermal transmittance, the building located in Castellón de la Plana (climate zone B3 according to CTE, meaning a mild winter and a hot summer) was considered. For each refurbishment solution, the minimum thermal insulation thickness to comply with the limit transmittance value set by the CTE for roofs that come into contact with outside air ($U_{LIM}$ = 0.44 W/m$^2$K) [10] was determined, and a commercially available thermal insulation thickness was assigned. As presented in Figure 3, all the refurbished systems' thermal transmittances fell within the 0.417 and 0.425 kW/m$^2$K range.

Besides determining the weight of each refurbishment solution, the expected overloads according to the regulations in force at the time of construction were also calculated [22,23]. The weight criterion indicated that the load-bearing capacity of the pre-existing roof permits an increase in overload of 0.5 kN/m$^2$. Consequently, the best refurbishment solution would be a roof with a gravel finish (NWGRA, 0.14 kN/m$^2$), and the worst would be a green roof (NWGREE, 2.12 kN/m$^2$), with the other solutions somewhere between these values. Despite green roofs' good energy performances [24,25], this solution was ruled out for rehabilitation due to its excessive weight. Although the solutions WINVR and WINVAD slightly exceeded the limitation, they were considered, but with some modifications to fulfill the weight requirement.

The cost of each refurbishment solution was quantified by taking the IVE price base [26] and some commercial solutions as references, and by adopting average values. The most economical solution was permeable paving at 58.31 EUR/m$^2$, and the most expensive one was the green roof at 92.89 EUR/m$^2$.

Figure 4 summarizes the adopted qualitative values (from 1 to 5, with the worst values in red and the best ones in green) according to the following criteria:

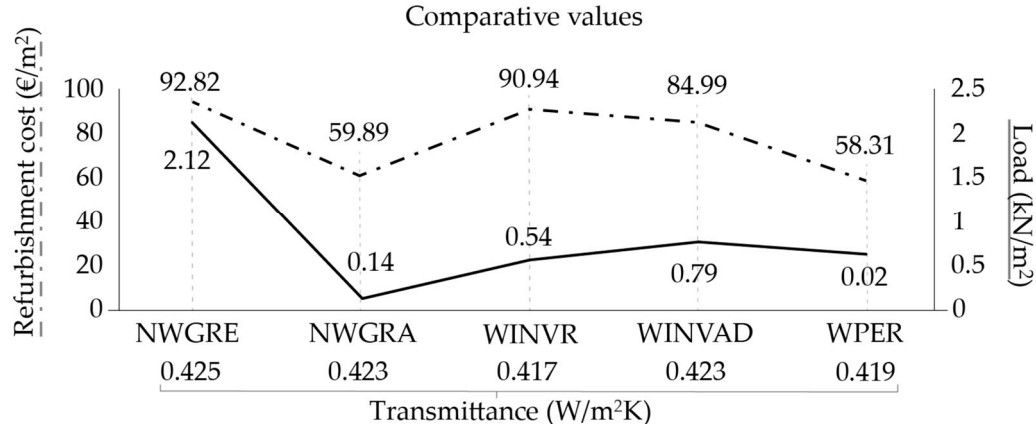

**Figure 3.** Thermal transmittance, load and refurbishment cost of the analyzed refurbishment solutions.

| | WINVR | WINVAD | WPER | NWGRE | NWGRA |
|---|---|---|---|---|---|
| A.1. Thermal insulation | 5.00 | 4.93 | 4.98 | 4.91 | 4.93 |
| A.2. Recovery—recycling | 5.00 | 1.00 | 5.00 | 4.00 | 5.00 |
| E.1. Initial investment cost | 3.21 | 3.43 | 5.00 | 3.14 | 4.87 |
| E.2. Maintenance | 5.00 | 3.00 | 5.00 | 3.00 | 5.00 |
| P.1. Ease of execution | 4.00 | 3.00 | 4.00 | 1.00 | 5.00 |
| P.2. Acoustic insulation | 4.00 | 4.00 | 4.00 | 5.00 | 4.00 |
| P.3. Weight of the system | 1.30 | 0.90 | 1.13 | 0.33 | 5.00 |
| P.4. Waterproofing—sealing | 5.00 | 5.00 | 5.00 | 5.00 | 5.00 |
| P.5. Aesthetic | 4.00 | 4.00 | 4.00 | 5.00 | 3.00 |
| **Mean values** | **4.06** | **3.25** | **4.23** | **3.49** | **4.64** |

Worst / Best

Tone range: 1 2 3 4 5

**Figure 4.** Assessment of the qualitative criteria for every refurbishment solution and score obtained before weighting criteria.

A.2 Recovery–recycling: The construction solution that meets the worst refurbishment is adhered paving. This is because this material requires applications with a cementitious adhesive, which makes it impossible to recover the parts to be reused. So, it was assigned the most unfavorable value of 1. This decision is supported by Fayos [27], who analyzed the environmental impact of flat roof construction solutions using the Life Cycle Assessment (LCA) methodology to conclude that the solutions with fixed flooring had the strongest environmental impact, followed by raised flooring and extensive green roofs. Finally, although the green roof was also recoverable, it was assigned a value of 4, which was somewhat lower than the previous ones due to the higher risk of degradation caused by vegetation, such as the waterproof membrane breaking due to roots.

E.2. Maintenance (durability–cost–periodicity): A score of 5 was assigned to the gravel, raised paving and permeable paving solutions because maintenance work on these rehabilitation solutions is minimal. The roofs with adhered paving may present damaged paving pieces due to the tensions generated on pieces given their installation with mortar

with green roofs. The main issue was the possible deterioration of any layer in the solution caused by vegetation.

P.1. Ease of execution: The most unfavorable was the green roof, with a score of 1, and the easiest to execute was the gravel roof, with a score of 5. The other two construction solutions were considered similar as regards this parameter, with a score of 4 (they are dry-laid solutions but require cutting and distributing pieces to adapt them to the roof).

P.2. Acoustic insulation: The best suitability was assigned to the green roof due to substrate and vegetation layers, with a score of 5. The other solutions proved similar in terms of composition and the system's mass, with differences in finishing. So, they were considered similar, with a score of 4.

P.4. Waterproofing–sealing: According to the assumption of correct execution, a score of 5 was assigned in all cases because all the solutions were perfectly watertight.

P.5. Aesthetics: The most favorable score was assigned to the green roof, with a score of 5, and the worst to the gravel roof, with a score of 3. The remaining solutions, with a wide range of aesthetic possibilities depending on the tiles selected for the roof's aesthetic finish, were assigned an intermediate score.

In prioritization terms, it was visually observed that the biggest number of favorable criteria corresponded to the gravel finish solution, followed by permeable paving, raised paving and adhered paving. The green roof had the most unfavorable criteria, whose weight was particularly important in rehabilitation. The adhered paving inverted roof obtained the lowest unweighted average value because most scores were medium or low.

3.3.2. Refurbishment Solutions Currently Used: Interviews with Contractors

A semistructured interview was held with active professionals in the construction sector to collect information on the specific roofing systems used in flat roof refurbishments. Interviews were conducted with six construction companies and professionals in the building renovation sector to seek information about the reality of the renovations that had been recently carried out. Interviewees answered the questions that appear below:

1. Identification of the work/project;
2. Location of the construction site (population);
3. Approximate age of the building/roof;
4. Approximate area of the roof ($m^2$);
5. Initial roof type;
6. Solution applied in renovation;
7. Approximate price of refurbishment (EUR/$m^2$) or overall budget (EUR).

In general, the professionals highlighted the fact that flat roof refurbishments exclusively aim to repair moisture problems, which originate in the waterproofing membrane and/or at singular points. In most cases, these actions are not used to incorporate thermal insulation or to improve buildings' energy efficiencies, seeing that users do not generally demand all this because this makes interventions more expensive. Moreover, they perceive that such an improvement only benefits the neighbors on the top floor, while the intervention is paid by the whole community of owners. These conclusions fall in line with the observation made by Ramos in his doctoral thesis [28], who observed that adding thermal insulation to roofs impacts the thermal comfort of top-floor dwellings in multifamily multistorey buildings. This fact may compromise the promotion of such measures when a community of owners makes consensual decisions when the economic criterion in refurbishment solution selections is usually the most important one.

So, even though current regulations have made an enormous effort to progressively improve buildings' energy efficiencies and to increase the proportion of buildings with almost zero energy use, their application is almost exclusively relegated to new construction and is optional in refurbishments. This leads to the question of what criteria are used for decision making when carrying out roof refurbishment. This aspect is what the following subsection seeks to clarify.

### 3.3.3. Survey on Weight Criteria in the Multicriteria Analysis

To estimate the weight of the criteria that can lead to a decision being made about the renovation of a building envelope, a questionnaire was carried out with two population groups: professionals related to the construction sector and homeowners. In both cases, the three above-assessed categories and indicators were used. Each indicator was assessed on a Likert scale, where 1 was considered the most unfavorable and 5 the most favorable. The normalized values were then obtained to evaluate the solutions.

A total of 27 responses were obtained from the professional sector and 55 from homeowners. The collected surveys included an equal number of responses from both men and women. The age range of the homeowners who participated in the survey was consistent with the proportion of homeowners in each age group, which explained the few respondents in their 20s. All the respondents in the expert surveys were professionals in the building sector, with 20 out of 27 respondents having more than 20 years of experience. The detailed breakdown of the survey respondents is as follows: 55 homeowners participated in the study, of whom 24 were women, 30 were men, and 1 person did not answer. The participants' age range was between 24 and 60 years old, with 4 in their 20s, 11 in their 30s, 22 in their 40s, 16 in their 50s and 2 in their 60s. Similarly, 27 experts participated in the survey, of whom 12 were women, 14 were men and 1 person did not answer. All the experts were from the architecture and building engineering field, and their age ranged from 27 to 84 years.

The calculation of the weighting applicable to each criterion was performed by determining the average score assigned by the survey participants to each criterion, and dividing this value by the sum of the average score of all the criteria, as presented in Equation (2). The results are collected in Table 3.

$$\%Pi = Vi / \sum (Vi) \times 100, \tag{2}$$

where:

Pi is the weighting factor for criterion i;
Vi is the value assigned to criterion i.

**Table 3.** Weighting coefficients.

| Indicators | Users | Experts |
|---|---|---|
| A.1. Thermal insulation | 13.5% | 12.7% |
| A.2. Recovery–recycling | 10.48% | 12.50% |
| E.1. Initial investment cost | 10.72% | 10.47% |
| E.2. Maintenance | 12.00% | 10.54% |
| P.1. Ease of execution | 8.90% | 10.15% |
| P.2. Acoustic insulation | 11.70% | 9.83% |
| P.3. Weight of the system | 9.46% | 10.68% |
| P.4. Waterproofing–sealing | 13.58% | 13.25% |
| P.5. Aesthetic | 9.68% | 9.87% |
| Total | 100% | 100% |

The values obtained from surveys were used as weighting factors to obtain a composite index, which allows the overall assessment of the suitability index (SI) of each proposed construction solution by considering all the criteria according to Equation (3):

$$SI = \left( \sum Pi \times Vi \right) / 100, \tag{3}$$

where:

SI is the sustainability index;
Pi is the weighting factor for criterion i;
Vi is the value assigned to criterion i.

As Figure 5 illustrates, all the criteria were generally rated above 3. Two criteria stood out: thermal insulation and watertightness. In contrast, aesthetics was perceived as the least relevant by both users and experts. One particularly noteworthy finding was the little importance that users attached to the system's weight compared to the value given by experts. This was a determining factor for the viability of the system's implementation from a structural stability point of view. Although values were similar, users rather than experts gave slightly higher figures to economic aspects, and it was the other way around for the aspects related to the recovery and recycling of material and to the ease of execution, which were better valued by experts. Figure 5 shows the multicriteria results when considering the weighting coefficients obtained from the corresponding questionnaires for users and experts, with SIu and SIe, respectively. As can be observed, the highest resulting values were for the non-walkable roofing system with gravel, and the lowest values were for the walkable roof finished with adhered paving.

| WINVR | | WINVAD | | WPER | | NWGRE | | NWGRA | |
|---|---|---|---|---|---|---|---|---|---|
| SIu | SIe | SIu | SIe | SIu | SIe | SIu | SIe | SIu | SIe |
| 4.15 | 4.12 | 3.38 | 3.28 | 4.33 | 4.29 | 3.65 | 3.55 | 4.67 | 4.68 |

**Figure 5.** Suitability index values obtained from users and experts (weighted average of criteria).

## 4. Stage 2 Results: Energy Performance for Refurbishment Solutions

For this approach, a specific urban area in the Grao neighborhood of Castellón, located on the coastline, was selected. In this area, a previous study by Pitarch et al. [29] was conducted, where the authors identified roof types and measured areas by obtaining surfaces by roof type. The main values are presented in Table 4. They are grouped in the area occupied by census sections 9001, 9002, 9003, 9004, 9005, 9006, 9007 and 9010, at the eastern end of the neighborhood, as presented in Figure 6a. The classification per construction period is summarized in Figure 6b, while Figure 6c depicts the identified roof types, which were sloped roofs, non-walkable flat roofs, walkable flat roofs and inner courtyards. Urban building energy modeling is assisted by Geographical Information System (GIS) maps to easily represent data in the territory [30]. Table 4 shows the distribution of areas per roof type, building typology (Sf—single-family; Mf—multifamily) and construction period. The total amount per type of roof and building typology is also provided. The areas of roofs to be considered in this study are highlighted in bold. They are roofs on high-rise buildings, being mostly flat roofs, which were built during the 1960–1979 period. These buildings were selected because approximately half the building stock in the Mf typologies was built during this construction period (see Figure 6b), when poor-performance energy solutions were employed.

**Table 4.** Area per building typology, roof type and year of construction, m$^2$.

| Time Period | 1840–1936 | | 1937–1959 | | 1960–1979 | | 1980–2006 | | 2007–2012 | | TOTAL |
|---|---|---|---|---|---|---|---|---|---|---|---|
| Typology | Mf | Sf | Mf | Sf | Mf | Sf | Mf | Sf | Mf | Sf | |
| Sloped | 2003 | 2622 | 452 | 771 | 612 | 466 | 18,717 | 10,293 | 4355 | 110 | 40,400 |
| Flat—non-walkable | 758 | 555 | 1281 | 786 | **5286** | 492 | 3245 | 2459 | 71 | 35 | 14,969 |
| Flat—walkable | 1237 | 1657 | 3379 | 1742 | **30,837** | 2271 | 16,392 | 1091 | 2498 | 127 | 61,231 |
| Inner Courtyard | 390 | 672 | 510 | 712 | 7609 | 650 | 14,119 | 8657 | 1993 | | 35,314 |
| TOTAL | 4388 | 5507 | 5622 | 4011 | 44,344 | 3878 | 52,474 | 22,501 | 8917 | 272 | 151,913 |

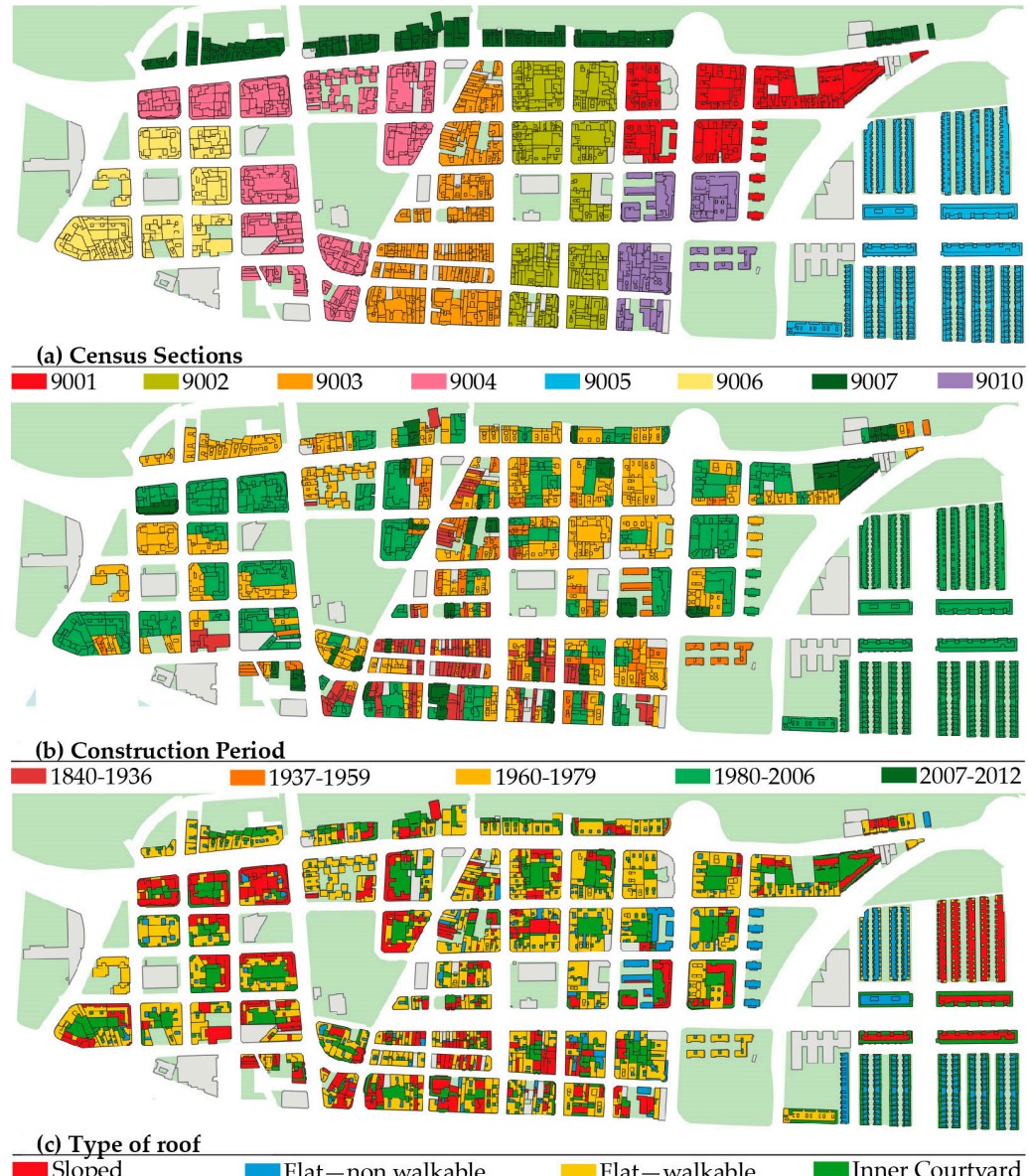

**Figure 6.** Geographical Information System maps representing the building stock in the area by (**a**) census sections, (**b**) construction period and (**c**) roof type.

To select a statistically representative building of the building stock in the urban area under study, the Cadastral data of the buildings included in the research area were collected following the process previously used by authors like Martín-Consuegra et al. [31]. For the buildings built during the selected construction period, the following variables were analyzed: the number of floors, the number of dwellings and the roof area. Some visits to the area were made to better collect data. A basic statistical analysis allowed the average number of floors, the number of dwellings and roof areas to be estimated for the buildings built in this area from 1960 to 1979, with the following results: 6.1 floors, 31 dwellings, 298.6 m$^2$ of roof area, 253.4 m$^2$ of walkable roof area and 36.7 m$^2$ of non-walkable roof area.

Most Mf buildings during this period present walkable roofs, while the small area of non-walkable roofs usually corresponds to the stairwell cover. To determine the energy rating with a simulation tool, an entire building has to be considered. Figure 7 shows the representative building of the selected studied neighborhood according to statistical criteria for this study stage. It provides a visual description of the most characteristic building typology in the neighborhood. It is located at 45 Alcocebre Street, was built in 1967 and

is located on a rectangular plot covering 262 m². It consists of a ground floor, five upper floors and three commercial premises on the ground floors. Its total surface area is 229 m², with 4 flats per floor, which totals 20 flats ranging from 64 to 79 m², with a total residential surface area of 1399 m².

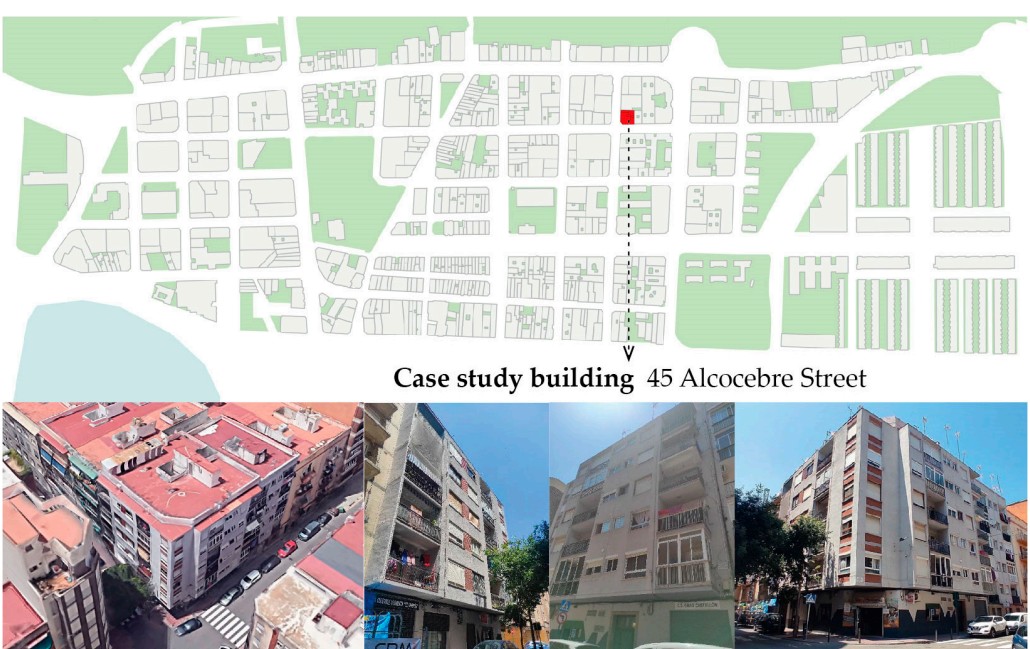

**Case study building** 45 Alcocebre Street

**Figure 7.** Case study building that is statistically representative of the neighborhood.

After considering the best refurbishment solution for walking roofs obtained in the previous stage by the multicriteria analysis with weighting criteria (WPER—walkable with permeable paving), the next step was to estimate an order of magnitude of energy savings by applying this solution in the urban area wherever the starting solution of the selected period was identified. To do so, the building geometry was checked by using cadastral cartography, the measurements taken at the site and by collecting data about existing constructive solutions. Next, the building was modeled using the CERMA v5.11 tool (Valencian Building Institute, IVE, and Asociación Técnica Española de Climatización y Refrigeración, ATECYR). This tool is one of those officially approved by the Spanish Ministry for Ecological Transition to certify buildings' energy performances (https://energia.gob.es/desarrollo/EficienciaEnergetica/CertificacionEnergetica/DocumentosReconocidos/Paginas/procedimientos-certificacion-proyecto-terminados.aspx, last accessed on 1 December 2023).

The inputs in software, as in other energy performance certification tools, are the climate data of the site, building typology, year of construction, orientation, composition of the thermal envelope and its main measures, together with the main information on building service facilities.

Simulation was conducted for three scenarios: first, the building in its original state (OR); second, the same building with the roof refurbishment by the WPER solution and appropriate thermal insulation to fulfill the requirements set out by the Technical Code for Building in its Energy Savings section (CTE-HE-1); and, finally, improving the thermal insulation of the WPER refurbishment solution by considering commercial formats for the thermal insulation of this permeable paving type available on the market (insulation thicknesses of 40, 50, 60 and 80 mm).

The original building's energy performance class was Class E on a scale from A to G, with emissions of 33.06 kgCO$_2$/m² per year. Despite poor insulation and the obsolescence of constructive systems and facilities, the rate is consistent with the climate zone and with mild winters. The study results are summarized in Table 5, where the values for emissions,

energy use and heating/cooling demands appear for each scenario, together with the reached rate and the percentage of savings when comparing the original solution to the refurbished one (in brackets).

**Table 5.** Simulation values and energy class in the representative building.

| | CO$_2$ Emissions (kgCO$_2$/m$^2$ Year) | Primary Energy Use (kWh/m$^2$ Year) | Heating Demand of Final Energy (kWh/m$^2$ Year) | Cooling Demand of Final Energy (kWh/m$^2$ Year) |
|---|---|---|---|---|
| Original | 33.06 E | 168.81 E | 84.10 G | 14.27 D |
| 40 mm | 31.22 E (−5.57%) | 159.93 E (−5.26%) | 77.81 G (−7.48%) | 12.97 C (−9.11%) |
| 50 mm | 31.14 E (−5.81%) | 159.55 E (−5.49%) | 77.54 G (−7.80%) | 12.91 C (−9.53%) |
| 60 mm | 31.11 E (−5.90%) | 159.41 E (−5.70%) | 77.44 G (−7.92%) | 12.89 C (−9.67%) |
| 80 mm | 31.02 E (−6.17%) | 158.96 E (−5.83%) | 77.13 G (−8.29%) | 12.82 C (−10.16%) |

As expected, fulfilling the CTE indicated scarce improvement in terms of carbon emissions or energy use when only the roof of a building is refurbished, which implied a decrease in these indicators of around 5–6%. This building's percentage is slightly higher than that published by Abdeen et al. [32], who obtained a 2.3% reduction by improving roof insulation. However, differences grew for heating/cooling demands, at 9% and 10%, respectively. When the tool simulated the commercial formats, the values lowered because insulation thickness increased, which is logical. As the price of paving in those cases could be the decisive factor for selection, the cost efficiency of the solution was analyzed to select the compromise solution [33,34]. To do so, the price of investing in refurbishment was estimated, together with the savings made in energy use. The optimal cost method was applied according to the Commission Delegated Regulation (DR; EU) No. 244/2012 of 16 January 2012, which supplements Directive 2010/31/EU of the European Parliament and the Council on buildings' energy performance. This method allows a comparative methodology framework to be established for calculating cost-optimal levels of minimum energy performance requirements for buildings and building elements. For the representative building, Equation (4) was applied:

$$C_{g(\tau)} = C_I + \sum_j \left[ \sum_{i=1}^{\tau} (C_{a,i}(j)R_d(i) + C_{c,i}(j)) - V_{f,\tau}(j) \right], \tag{4}$$

where $C_g(\tau)$ is the global cost (referring to the starting year $\tau$ 0) over the calculation period to be estimated. Table 6 presents the meaning of each term in Equation (4), as well as the values and starting hypothesis adopted in the analyzed case:

**Table 6.** Definition of the terms in Equation (4), values and the starting hypothesis.

| Term | Definition | Hypothesis |
|---|---|---|
| $\tau$ | Calculation period. | 30 years (as proposed by the DR for residential use). |
| $C_I(j)$ | Initial investment costs for measure or set of measures j. | Obtained using the unit price from the Valencian Building Institute, IVE, construction database (https://bdc.f-ive.es/BDC22/1, accessed on 1 December 2023): EUR 41,373.43, EUR 42,381.84, EUR 45,095.45 and EUR 45,509.18 for 40, 40, 60 and 80 mm, respectively. |
| $C_{a,i}(j)$ | Annual cost during year i for measure or set of measures j. | Energy saving due to the solution by adopting an average energy price of 0.3 EUR/kWh: EUR 3726.94, EUR 3886.42, EUR 3945.18 and EUR 4134.05 for 40, 40, 60 and 80 mm, respectively. |
| $C_{ci}(j)$ | Carbon cost during year i for measure or set of measures j. | A total of 0 EUR. This term is implemented from a macroeconomic perspective. Spain selected the financial perspective. |
| $R_d(p)$ | $R_d(p) = \left( \frac{1}{1+\frac{r}{100}} \right)^P$ is the discount factor for year i based on discount rate r, where p means the number of years from the starting period. | Three rate factors *r* to undertake a sensitive analysis: 1%, 4% and 6% for the optimistic, medium and pessimistic scenarios, respectively. |
| $V_{f,\tau}(j)$ | Residual value of measure or set of measures j at the end of the calculation period (discounted to starting year $\tau$ 0). | A total of 0 EUR. No residual value is considered for the selected solutions. |

Table 7 shows the global cost, which considers a net present value of the investment and the payback period (Pp) of the different considered solutions. The last row indicates an additional scenario for the 80 mm solution by assuming an annual increase of 2% in the energy price, which is likely to progressively rise.

**Table 7.** The global cost and payback period (Pp) of the refurbishment solutions considering several scenarios for the sensitive analysis.

| Solution | r1% | | r4% | | r6% | |
|---|---|---|---|---|---|---|
| | Cg($\tau$) (EUR) | Pp (Years) | Cg($\tau$) (EUR) | Pp (Years) | Cg($\tau$) (EUR) | Pp (Years) |
| **40 mm** | 53,181.74 | 12 | 23,920.24 | 14 | 11,644.68 | 17 |
| **50 mm** | 56,330.30 | 11 | 25,779.87 | 14 | 12,963.20 | 17 |
| **60 mm** | 55,147.32 | 12 | 24,121.99 | 15 | 11,105.94 | 18 |
| **80 mm** | 59,657.63 | 11 | 27,105.84 | 14 | 13,448.96 | 17 |
| **80 mm + 2%** | 95,488.16 | 10 | 47,674.82 | 12 | 28,132.29 | 14 |

Like previous studies, this one used basic statistics to simulate the building stock [33–35]. The magnitude of the potential savings in the neighborhood was estimated by considering the total area of the flat roofs of this building typology and the construction period (see Figure 6 and Table 4). Accordingly, the refurbishment of 36,123 m$^2$ of the considered roof with the WPER solution of 80 mm thickness would save 339,682.36 kgCO$_2$ and 1,640,132.98 kWh per year. This estimation was obtained by the theoretical simulation of a representative building. Consequently, it must be taken as an order of magnitude rather than as an accurate amount. However, in economic terms, with an investment of EUR 5,416,567.08, and considering the used energy cost, it would lead to savings of EUR 492,039.89 in energy terms. This would imply an estimated return of investment of 11–17 years, depending on the tax rate scenario. This Pp could be reduced to 10–14 years when assuming a scenario with an annual increase in the energy price of 2%. As observed by Jaber [36], these relatively long Pps make the sole rehabilitation of building roofs unattractive from an economic point of view.

## 5. Discussion

After analyzing the different proposed rehabilitation solutions, and in view of their weaknesses, distinct strategies can be defined to achieve more efficient solutions. Among them, several possibilities can be studied:

- Study the possibility of developing lighter rehabilitation systems by using thin ceramic tiles or sheets. This strategy can be employed with raised flooring by applying a reinforcement layer in direct cladding and in permeable flooring to allow water permeability in tile joints;
- Raise public awareness about the importance of energy efficiency in buildings so that renovations are put to the best possible use to incorporate thermal insulation. This could be performed with existing financial aid so that homeowners' associations could consider it. The cost of the intervention that incorporates thermal insulation is similar or even lower than simply repairing existing damp problems;
- Study the possibility of using recycled aggregate gravel as protection for inverted roofs, which would increase the overall sustainability of the solution.

After analyzing some potential scenarios, possible future considerations can be the following:

- Subsidies and public support for interventions should be maintained to reach an efficient building stock, bearing in mind that low-performance building owners usually face more difficulties that involve making high investments;

- It is necessary to raise more long-term awareness by considering a more holistic view and realizing the benefits of prolonging buildings' life spans through proper maintenance and upgrading;
- Building roofs are normally underrated, perhaps because they cannot be seen from the street level. However, they are important elements in the envelope and are crucial for prolonging the life span of buildings and for promoting proper building functionality. Their proper maintenance will surely avoid future costly repairs.

This work has presented a limited number of refurbishment solutions by basing their environmental performance on improving thermal insulation, which allows the thermal transmittance of the envelope to be reduced. However, this study has not explored other rehabilitation options like those mentioned by Madushika et al. [37], i.e., using paints or highly reflective clay tiles, which could complement the analyzed solutions.

## 6. Conclusions

This research work assesses the viability of various roof rehabilitation systems by determining the best option based on a multicriteria analysis. To determine the potential energy improvement, the best solutions are simulated in a representative building of a neighborhood located in Castellón de la Plana (East Spain). The energy improvement possibilities at the neighborhood level are determined by extrapolating these refurbishment proposals to the study area. Based on the obtained results, the following conclusions can be drawn:

- The multicriteria assessment indicates the gravel system as the most favorable one when considering the A, E and P aspects together. Raised paving and permeable paving systems obtain an intermediate overall rating, which can be improved by reducing the system's weight or the cost of raised floors;
- Despite the importance of thermally insulating the building envelope and improving the building stock's energy efficiency, the current regulation (DB HE1) is only applicable in certain refurbishment cases to obtain buildings with almost zero energy use. So, owners tend to look for the cheapest solution;
- Users attach more importance to the cost of the investment, and this factor sometimes determines the feasibility of refurbishment. As highlighted by renovation professionals, thermal insulation is generally not incorporated into refurbishments of flat roofs of Mf housing buildings because it makes refurbishment more expensive and is considered to benefit only the top-floor dwellings;
- The fact that roof renovation solutions are overweight is extremely important and conditions their application. However, their importance is not perceived as such by users and experts;
- Although green roofs are highly desirable from a sustainability point of view, they should be ruled out as a refurbishment solution for existing buildings for being overweight;
- Roof refurbishment's cost efficiency does not seem very optimistic. Investments of about 60–90 EUR/m$^2$ are needed, which means slightly improved energy performance, with savings in overall non-renewable primary energy use of around 5% or 6% for a benign climate zone, such as that herein analyzed. However, considering that this refurbishment means acting on 20% of the building's total thermal envelope, the result is not so low;
- The Pp for the studied cases is 10 years in the best-case scenario and 18 years in the worst-case scenario;
- It is important to emphasize the fact that not only roof insulation is improved with refurbishment but also waterproofing and, therefore, habitability conditions, especially those of the top-floor dwellings. This argument reinforces the convenience of roof refurbishment because it not only improves the building's energy performance but also increases its life span by improving and repairing a very exposed area of the

building envelope, which avoids potential diseases. All these reasons reinforce the alignment of roof refurbishment with buildings' sustainability;

- If the results obtained for the statistically representative building are extrapolated to the neighborhood scale, the total values are quantitatively significant, with an annual saving in atmospheric emissions of almost 340 tons of $CO_2$, and a reduction in energy use of over 1.6 million kWh. Thus, the potential improvement in the area is quite high.

In the last two decades, there has been a significant increase in the number of scientific studies that explore ways to improve buildings' energy efficiencies. This has been conducted in response to evolving regulations and a global push toward decarbonization of the economy. The new EPBD proposal includes important aspects that provide insights into how sustainability can be promoted effectively in the building industry. Some of these include more ambitious energy performance requirements for both new and renovated buildings, the consideration of carbon emissions throughout a building's entire life cycle, the adoption of "staged renovation" as a solution to the high upfront costs and financial mechanisms to boost the old building stock's renovation. These aspects show the future trend in the building sector, and highlight the importance of roof refurbishment in enhancing a building's durability and overall performance.

**Author Contributions:** M.J.R., methodology; all authors, formal analysis; M.J.R. and L.R., investigation; all authors, data curation; Á.M.P. and I.A., supervision; L.R., writing—original draft preparation; M.J.R. and L.R., writing—review and editing. All authors have read and agreed to the published version of the manuscript.

**Funding:** This work was funded by Projects "Rooftiles and Rooftiles II" with a collaboration contract with the research projects with references IMDEEA/2021/34 and IMDEEA/2022/7, respectively, of the Ceramic Industries Research Association (AICE—Asociación de Investigación de las Industrias Cerámicas), in 2022 and 2023. These projects were supported by the Instituto Valenciano de Competitividad Empresarial (IVACE) through FEDER (Fondos Europeos de Desarrollo Regional) funding.

**Institutional Review Board Statement:** This study did not require ethical approval.

**Informed Consent Statement:** Informed consent was obtained from all subjects involved in this study.

**Data Availability Statement:** All data from buildings are public and were retrieved from the Cadastral Office. All respondents to the surveys were informed about the purpose of this study and signed an informed consent form; the responses were anonymized and stored for research purposes, with access restricted to the study's authors.

**Conflicts of Interest:** The authors declare no conflicts of interest.

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
