# Peer review of "A Roof Refurbishment Strategy to Improve the Sustainability of Building Stock: A Case Study"

_sustainability, doi:10.3390/su16052028_

Round 1
Reviewer 1 Report
Comments and Suggestions for Authors
In the paper the Authors undertakes interesting and important (especially nowadays) topic referring to roof refurbishment strategy in the context of improvement of sustainability of residential, multifamily buildings. The research presented in the article covers environmental, economic and performance factors to select the most efficient and sustainable solutions of roof refurbishment. To achieve the aim of the paper Authors conducted interviews with construction professionals, and a survey of experts and homeowners. The Five solutions were selected and examined to estimate energy savings, payback periods and the environmental impact for a representative building of the study area. The Authors point at the best solution based on selected criteria. As mentioned above topic is very interesting and up to date, however Authors should consider the minor suggestions listed below:
· The aim of the work should be exposed and highlighted in the abstract
· The authors should consider expanding the information in the introduction regarding the "fit for 55" regulation and in particular Energy performance of buildings in the part regarding - existing buildings should be transformed into zero-emission buildings by 2050
· Even though the research was properly designed, it is necessary to consider whether an appropriate number of surveys were conducted to make the research representative. Therefore, the authors should answer how many construction experts and other of users/owners of buildings were surveyed?
Author Response
Dear reviewer,
The authors express their gratitude to the reviewers for their valuable comments and observations on the paper. All issues raised have been addressed and the authors believe that the changes made have considerably enhanced the quality of the paper. The changes made to the manuscript have been clearly indicated using the word's change control function. The authors' responses to the reviewers' comments can be found in the Reviewer's comments with responses file, where the authors' responses are in blue and the newly added text is black, between quotation marks.

Reviewer 2 Report
Comments and Suggestions for Authors
- Figures should have more detailed labels for better understanding.
- Avoid starting sections with phrases like "Figure 2 summarizes the process followed to undertake this project." Instead, provide a direct and concise description of the figure's content.
- Adjust the font size of figures and tables to make them more visually appealing and consistent with the rest of the document.
- Consider dividing the "Results" section into two subsections: "Analysis" and "Discussion and Results" for better organization.
- Rewrite the section "Refurbishment solutions currently used: interviews with contractors" to exclude unnecessary questions and focus on relevant information.
- Clarify the purpose of Figure 7.
- Remove the URL from Table 6 and include it as a footnote instead.
- Provide details on how the equations were validated and tested.
- Merge the last two sections into one titled "Conclusions" and use numbered points instead of bullet points for clarity.
Author Response

(The authors gave the same response as above.)

Reviewer 3 Report
Comments and Suggestions for Authors
It is a very useful contribution to decision-making on the subject. The images of both graphics, maps and photographs are expressive; Only the maps of the neighbourhood (Figure 6) are small, although their greater detail is not essential for the purpose of this article.
Author Response

(The authors gave the same response as above.)

Reviewer 4 Report
Comments and Suggestions for Authors
The theme is interesting; it would have been desirable the extension of the study to the façade of the buildings of the neighbourhood, which I recommend for future research.
Author Response

(The authors gave the same response as above.)

Reviewer 5 Report
Comments and Suggestions for Authors
The manuscript addresses the impact of roof renovation on improving energy efficiency in flat-roofed multifamily dwellings in medium-sized Mediterranean cities. A multi-criteria analytical model in the form of a questionnaire considering environmental, economic and performance factors was constructed to select the best roof renovation solution. The results of the study have some practical applications. However, there are still the following issues that need to be improved.
1、In proposing solutions, the authors do not seem to make it clear on which participant's position the analysis is based. Since different participants have different interests in the process, there is a need to clarify the position from which the proposal is made.
2、In the references section, authors are advised to make more references to newer and higher quality literature to enhance the persuasiveness of their arguments.
3、The manuscript states that author weights were determined based on 27 responses from the professional sector and 55 from homeowners. The number of questionnaires collected appears to be small, and also the distribution of the specific survey population was not analysed to determine whether the selection of the survey population was justified. The reasonableness of the selection of that population, in turn, would have a greater impact on the reliability of the results.
4、It is recommended that the authors, in their conclusions, strengthen the integration with the analyses in the manuscript. At the moment it seems that the results derived from the conclusions are not sufficiently integrated with the main body of the manuscript.
5、It is recommended that in the introduction, the authors include a section to better express the importance of renovating and remodelling roofs.
Author Response

(The authors gave the same response as above.)

Reviewer 6 Report
Comments and Suggestions for Authors
This study focuses on roof refurbishment in Mediterranean climates, addressing a gap in sustainability and building efficiency literature. Its specificity extends its applicability to similar regions. The comprehensive methodological approach, combining multicriteria analysis with stakeholder interviews and simulation tools, ensures a nuanced understanding of the topic. This mixed-methods strategy integrates quantitative data on energy savings and environmental impacts with qualitative insights into stakeholder preferences and decision-making criteria. The study's engagement with recent sources aligns with current debates and strengthens its contributions to both theoretical and practical aspects of sustainable building practices.
Author Response

(The authors gave the same response as above.)
